A consumer emotion analysis system based on support vector regression model

Huo Mingkui huomingkui@cust.edu.cn
Li Jing
Changchun University of Science and Technology, School of Economics and Management , Changchun , Jilin , China
Jiang Yizhang
Electronic publication date: 2023 May 9
Publication date: 2023
Volume: 9
Electronic Location ID: e1381
Received 2023 Feb 3; Accepted 2023 Apr 13
Copyright: ©2023 Huo and Li
Copyright year: 2023
Copyright holder: Huo and Li
License: This is an open access article distributed under the terms of the Creative Commons Attribution License, which permits unrestricted use, distribution, reproduction and adaptation in any medium and for any purpose provided that it is properly attributed. For attribution, the original author(s), title, publication source (PeerJ Computer Science) and either DOI or URL of the article must be cited.
License URL: https://creativecommons.org/licenses/by/4.0/

Keywords: Emotional analysis, Willingness to pay, SVR, RCNN

Funding: The authors received no funding for this work.

==============================
The effective means to stimulate economic growth is to enhance consumers’ consumption capacity. Because many consumers have different consumption habits, they will pay different attention to products. Even the same consumer will have different shopping experiences when buying the same product at different times. By mining the online comments of consumers on the online fitness platform, we can find the characteristics of fitness projects that consumers care about. Analyzing consumers’ emotional tendencies towards the characteristics of fitness programs will help online fitness platforms adjust the quality and service direction of fitness programs in a timely manner. At the same time, it can also provide purchase advice and suggestions for other consumers. Based on this goal, this study uses an optimized support vector regression (SVR) model to build a consumer sentiment analysis system, so as to predict the consumer’s willingness to pay. The optimized SVR model uses the region convolution neural network (RCNN) to extract features from the dataset, and uses feature data to train the SVR model. The experimental results show that the SVR model optimized by RCNN is more accurate. The improvement of the accuracy of consumer sentiment analysis can accurately help businesses promote and publicize, and increase sales. On the other hand, the increase in the accuracy of emotion analysis can also help users quickly locate their favorite fitness projects, saving browsing time. To sum up, the emotional analysis system for consumers in this paper has good practical value.

Introduction

E-commerce sites have become widely used as the Internet has grown in popularity. People’s lives now routinely include making purchases online. Online shopping offers a plethora of options, but the quality of these stores and their wares varies widely. The primary objective of any business is to generate as much revenue as possible through the sale of its goods and services. Consumers will have a positive shopping experience if they are given the freedom to select the products they desire at the lowest possible price and in the shortest amount of time. The foregoing issues can be resolved by conducting a thorough analysis of consumers’ feelings (Kim et al., 2022; An et al., 2021). By shopping online, customers can track their browsing history, how long they spend on specific products, their purchase history, and how they rate various products. The vast majority of such information consists of textual content. Because of this, customer sentiment can be mined using data mining techniques (Sohail et al., 2021; Cudjoe & Cawooda, 2022; Halim, Ali & Khan, 2021). Emotion word selection in text, which is used in the emotion categorization approach, is tough to operate. It is challenging to construct an exhaustive and accurate emotional vocabulary due to the wide variation in emotional terminology used throughout situations. The construction of a more precise feature dictionary is thus essential for the research of various items and this review. Furthermore, there are substantial variations in the manner of expressing emotions amongst nations due to their distinct histories and cultural norms. Because of this, determining the true emotional state of a product is often a challenge.

To do an emotional orientation analysis means to conduct an analysis of the text, which involves the feelings and personal experiences of the customers. Emotion dictionary (Cochrane et al., 2022; Jang, Choi & Kim, 2022), machine learning (Kalaiarasi & Maheswari, 2021; Vadhnani & Singh, 2022; Yalsavar et al., 2022), and deep learning (Strawn, 2022; Stultz, 2021; Paguada et al., 2022) are some of the technologies that are frequently utilized in the numerous research projects that are conducted on text emotion analysis both in the United States and internationally. Alswaidan & Menai (2020) examines and compares four distinct approaches to recognizing the emotions conveyed in written text: a rule-based approach, a classical learning approach, a depth learning approach, and a hybrid approach. Researchers are able to have a better understanding of which method is more appropriate for which sort of data if they first evaluate the benefits and drawbacks of each of these methods. In Schwering et al. (2021), books are used as experimental data, and corpora are used to anticipate the feelings that characters in novels will experience. The findings indicate that extensive experience reading novels over time can effectively identify the emotions associated with an activity. Deep learning techniques are utilized in the aforementioned piece of scholarly literature (Braunschweiler et al., 2022). Mixed data contains audio and text data. The findings of the experiments indicate that the combination of speech and text has a greater effect on emotion recognition than either one alone. The research presented above is primarily applied to the task of emotion recognition in text at various levels of technical sophistication. In this work, we focus on the consumer sentiment of users of online fitness platforms and introduce technology that employs deep learning and machine learning to forecast the consumer sentiment of platform users.

Users of the online fitness platform get access to time-sensitive and easily accessible fitness resources and information. On the site, users have the ability to acquire a variety of services, including personal course customization, fitness punch reminders, fitness knowledge development, fitness experience engagement, and more. A significant number of people who are interested in fitness and people who have fitness-related demands frequent fitness platforms. Although the number of people using the fitness platform is constantly increasing, the percentage of users who actually pay for the service is rather low, and paid consumption is not particularly common. The percentage of permanent users and contributors is relatively low, whereas the majority of users are mobile or just passing through. The goal of numerous fitness platforms is to continually attract and keep customers by providing high-quality fitness courses and considerate fitness services; yet, the result of these efforts is not immediately apparent due to overlapping functionalities and content. The social function of the platform, particularly the function of recruiting users, fostering user habits, and creating trust through information exchange, has been neglected by the platform operation. Therefore, in order to better guide the operation of the online fitness platform and further enhance the user stickiness and willingness to pay, this paper introduces an improved SVR model to predict the user’s consumption emotion based on the use data of the online fitness platform. This model is designed to predict the user’s consumption emotion in order to further enhance the user stickiness and willingness to pay. This article primarily focuses on two different points. To begin, the SVR model is modified in order to enhance the accuracy of its prediction. Second, the enhanced SVR was used to do emotional analysis on the platform’s users, which resulted in an increase in the platform’s overall operating efficiency.

Related Research and Basic Theory

Definition of relevant concepts

The term “consumer” refers to individuals as well as households that buy and make use of a variety of consumer goods or services. Consumers are persons or organizations that buy and make use of a wide variety of goods and services. The term “consumer” can be used in a general sense.

The term “consumer behavior” refers to the dynamic process in which the consumer’s preferences, cognition, actions, and the surrounding environment all interact with one another in their day-to-day dealings. The following three aspects of consumer behavior are highlighted by this definition: (1) It is a process that is constantly changing; (2) It involves the interaction of a person’s preferences, cognition, behavior, and the environment; and (3) It involves transactions.

The term “consumer decision making” refers to the process in which customers thoughtfully consider the merits of a certain product, brand, or service and then settle on a course of action based on their findings. To purchase a particular item at the lowest possible price is one example of this approach.

The term “consumption emotion” refers to the psychological propensity of customers to make a purchase after browsing a particular product. It is also possible to think of it as the likelihood that customers are willing to make a purchase. When consumers have a particularly strong desire to buy something, they believe that the “consumption emotion” is at its peak. People have a tendency to believe that consumer sentiment is at an all-time low when they do not want to make excessive purchases.

Theory of consumer behavior intention

The term “willingness” refers to the people’ self-perceived likelihood of participating in particular behaviors (Murphy et al., 2021). As a result of the research that was done on the idea of will, a number of academics came up with the theory of rational behavior (Snippe, Peters & Kok , 2021). Figure 1 illustrates the core concept behind this theoretical framework. According to this theory, the likelihood of an individual to engage in a particular activity is dependent on the individual’s willingness to engage in that behavior. This willingness is jointly controlled by an individual’s attitudes and their own subjective norms. The level to which an individual is willing to engage in a particular behavior is a metric that can be used to assess intensity.

Figure 1 Rational behavior theory.

The rational behavior theory is an overarching framework for researching the elements that have an effect on an individual’s actions. It is possible to use the model of behavioral will and actual behavior that was constructed by applying this theory in order to explain the causal relationship that exists between the influence that certain beliefs, motivations, and other factors held by individuals have on behavioral will and actual behavior. This research investigates the effect that data from online reviews have on the emotional state of consumers, which is essentially an investigation into the connection between the emotional state of consumers and their purchasing decisions. As a result, the construction of this study model has a theoretical foundation thanks to the idea of rational conduct.

The theory of rational behavior has not only been widely implemented in a variety of contexts, but it has also sparked more theoretical investigation into the limitations of its scope. As a result, Wong & White (2022) present the theory of planned conduct by combining the idea of numerous attributes with the theory of rational behavior. The fundamental idea behind this hypothesis is depicted in Fig. 2.

Figure 2 Planned behavior theory.

The notion of planned behavior was developed with the intention of explaining the variations in spontaneous conduct. The idea of planned behavior asserts that an individual’s behavior intention is influenced not only by their behavior attitude and the subjective norms that they adhere to, but also by the behavior control that they engage in. The theory of planned conduct has been extensively used in a variety of contexts, which has paved the way for research on the development of follow-up attitudes by providing a solid framework.

Data characteristics of online fitness platform

Information interaction is a two-way process of information sharing and exchange based on user experience, the core of which is to meet users’ information needs and enhance user experience. Information interaction on online fitness platforms refers to the process of information exchange and feedback between users and platforms or between users and fitness influencers, which includes experience exchange, discussion, service consultation, service purchase and other contents. Based on different participants, interactions can be categorized into three types: human-machine interaction, human–human interaction and group interaction. Human–machine interaction indicates the interaction between users and systems, that is, when users search or consult on the platform, AI will provide automated communication services; human–human interaction happens when users communicate with fitness influencers or administrators on the platform regarding consultation over the contents or seek for personalized services; group interaction generally includes experience exchange, emotion interaction, discussion activities and other forms of interaction among users, fitness influencers, administrators and other participants, be it in public or in private forms. Users can obtain a sense of fulfillment in emotional terms as well as a sense of self-approval after participating in group interaction, which means group interaction embodies significant social features.

The information interaction of online fitness platform is characterized by professionalism, interest, real-time and personalization. Professionalism means that the information service subject relies on its information resource advantages to accurately analyze the needs of users and provide users with professional online fitness information services. This kind of professionalism contains the knowledge, experience and value characteristics of the information service subject, and transfers the positive fitness guidance based on the mainstream social values. Interest is an important influencing factor for users to participate in online activities. Through information exchange and interaction with other users and fitness bloggers, it can not only meet their own information needs, but also bring interesting experience in interaction. The information content in the platform includes the text content, video courses, pictures, etc., created by the fitness blogger, which are from the publisher’s own creation and have a distinctive personal style. Information content can bring interesting experience to users. In addition to daily clocking, users’ dynamic circle content also includes their personal sharing, which reflects their personal feelings and preferences. Users can bring joy to other users while entertaining themselves. Real time means that the platform can give high-speed response when users obtain information on the platform. Users can obtain information through the platform, communicate with other users online, feed back the use of the platform, evaluate the platform, update the information according to user needs, innovate the service content, and bring users a high-level use experience. The real-time nature of information interaction brings users the ultimate experience of rapid information acquisition, which directly affects their use satisfaction. Personalization is reflected in that the platform can customize fitness services for users, or recommend content for users according to their needs. The platform will recommend relevant information for different users according to their differences. In the process of information interaction, users can feel the personalization of information services, reflecting the characteristics of taking user needs as the core.

Consumption emotion prediction model

RCNN

Feature extraction is the primary use of CNN. A regression convolution neural network model structure (RCNN) similar to the one depicted in Fig. 3 is presented as a means of applying CNN to challenges involving regression. The input layer, the convolution layer, the pooling layer, and the fully connected regression layer are the components that make up RCNN. In this particular network, the processes that are primarily included are feature extraction and prediction. Two convolution layers, two maximum pooling layers, one activation layer, and one standardization layer are utilized in the process of feature extraction. The complete link layer and the regression layer are the components that make up the prediction process. Through a series of convolution and pooling layers stacked on top of one another, this model primarily collects features from the raw data that is fed into it. Through the utilization of the full connection regression layer, the characteristic data may accurately forecast the consumers’ intentions to make a purchase.

Figure 3 RCNN structure.

SVR

A significant use of SVM is known as SVR. They are both operating under the assumption that the regression hyperplane that lies closest to the data point is the most accurate one. However, different mathematical models can be expressed in a variety of different ways. The following describes the hyperplane of SVR: (1) fx=wTx+b.

The SVR model is: (2) minw,b12w2+γ∑i=1mIηfxi−yi.

Where γ is the regularization constant (γ>0), Iη is the insensitive loss function, and its function expression is: (3) Iη=0,z≤ηz−η,z >η

By introducing relaxation variables ϕi and ϕ ˆi, Eq. (2) can be expressed as: (4) min12w2+γ∑i=1mϕi−ϕ ˆis.t.fxi−yi≤η+ϕ ˆyi−fxi≤η+ϕ ˆϕ ˆ≥0,ϕ ˆi≥0,i=1,3,…,m.

The Lagrangian multiplier μi≥0,μ ˆi≥0,δi≥0,δ ˆi≥0 is introduced, and the Lagrangian function can be obtained from the Lagrangian multiplier method as follows: (5) L=12w2+γ∑i=1mϕi+ϕ ˆi− ∑i=1mμiϕi+ ∑i=1mδifxi−yi−η−ϕi+ ∑i=1mδ ˆiyi−fxi−η−ϕ ˆi

Substitute Eq. (1) into Eq. (5), let the partial derivative of L to w,b,ϕi,ϕ ˆi be 0, and get: (6) w= ∑i=1mδi−δ ˆixi

(7) ∑i=1mδi−δ ˆi=0

(8) δi+μi=γ

(9) δ ˆi+μ ˆi=γ.

Substitute Eq. (6)–Eq. (9) into Eq. (5) to get the dual problem of SVR: (10) max∑i=1myiδ ˆi−δi−ηδ ˆi+δi−12 ∑i=1m ∑j=1mδ ˆi−δiδ ˆj−δjxiTxjs.t.∑i=1mδ ˆi−δiδi≥0,δ ˆi≤γ

The procedure described above must adhere to the KKT criteria, which may be specifically translated into the solution formula that is as follows: (11) δifxi−yi−η−ϕi=0δ ˆifxi−yi−η−ϕ ˆi=0δiδ ˆi=0,ϕiϕ ˆi=0γ−δiϕ=0,

Substitute Eq. (10) into Eq. (1), and the SVR problem is: (12) fx= ∑i=1mδi−δ ˆixiTx+b

Where (13) b=yi+η−∑i=1mδi−δ ˆixiTx

Model construction

When it comes to predicting the feelings that customers have when making a purchase, RCNN and SVR models do not have very good prediction accuracy. The strength of RCNN’s capacity to extract features is one of its primary advantages, while the strength of SVR’s ability to do regression on nonlinear data is one of its primary advantages. As a result of this, the authors of this work propose improving the SVR prediction model by integrating the two different methods. The goal of this model is to train an SVR model using feature data that was extracted from the online data of customers using RCNN. The extracted feature data was then used as training data for the SVR model. The outcome of the SVR model is the result of a forecast of the customer sentiment. Figure 4 illustrates the modified SVR model’s structural components.

The RCNN feature extraction layer and the SVR classifier layer make up the essential components of the upgraded SVR model structure. The RCNN network included in this model has been somewhat modified in order to make the suggested model more effective. As a result, this model’s RCNN network looks a little bit different from the standard RCNN network. This model employs a total of three convolution layers in addition to a maximum pooling layer. This is due to the fact that a single RCNN network only has a limited number of convolution layers. A single RCNN model has its whole connection layer and regression layer removed, and an SVR classifier is put in its place. Feed the data into the neural network, and then perform the convolution pooling procedure multiple times in succession. RCNN is capable of carrying out spatial local feature extraction of feature factors right up until the point where the final pooling layer outputs the feature map. Then, in order to train the SVR classifier, the result features that were retrieved from the RCNN network layer are fed into the SVR classifier. In conclusion, an SVR classifier is applied in order to forecast the feelings of customers.

The enhanced SVR model is primarily composed of two components, which are feature extraction and prediction. Figure 5 illustrates the model’s complete operational flow as follows:

Figure 4 Improved SVR model structure.

Figure 5 Consumer sentiment prediction process.

The steps involved in the implementation of the prediction model that was utilized for this paper are as follows, according to the flow chart that can be found in Fig. 5:

Step 1: Data collection. Collect relevant data such as consumer browsing products, consulting products, purchase records and comments.

Step 2: Data processing. Create a standard for the data relating to the historical purchasing behavior of customers.

Step 3: After being preprocessed, the data set is then separated into the training set and the test set. The training data set consists of sixty percent of the total data set, while the test data set uses the remaining forty percent.

Step 4: Extraction of features using RCNN In the RCNN network, the core size of Convl, Conv2, and Conv3 are respectively 20, 25, and 50, and the step size is 1 ×1. After that, the maximum pool layer is used to lower the total amount of feature data, and the correlation features that exist between the input data and the output power are extracted. The largest pool tier has steps that are 2 squares on each side. Cut down on the amount of time spent computing over the network. In conclusion, the characteristic scale that satisfies the criterion was successfully obtained.

Step 5: Build up the SVR classifier. Following the training of the SVR classifier using the feature data obtained from the RCNN network, the results of the predictions are output.

Experimental Results and Analysis

Experimental data

This post makes use of the data set that was provided for the Google Analytics Customer Revenue Prediction competition that was hosted on the Kaggle website around September of 2018. By doing an analysis of the customer data set held by Google Merchandise Store, the goal of this competition is to see how accurately one can forecast the future spending power of individual customers. By utilizing this dataset, it is possible to confirm the originality and accuracy of the data source, in addition to demonstrating that the methodology presented in this study has value for practical application.

It is necessary to preprocess the data set in order to satisfy the requirements of the prediction model. This is due to the fact that the data set cannot be retrieved straight from the Google platform. The data set that was obtained directly comes from a session and functions as a sample vector. The user acts as the sample vector and is therefore the processing object of the model. As a result, it is essential to do preliminary processing on the customer data set from the Google Store using the session as the sample vector. The data set that was utilized for this study was the data set after it had been preprocessed. 1,653,827 different training sets were utilized in the experiment, while 398,762 different test data sets were utilised.

Experimental analysis

The mean absolute error (MAE), the mean square error (MSE), and the mean absolute percentage error (MAPE) are the primary evaluation metrics that are utilized in the process of analyzing the effectiveness of the prediction models that are put into use (MAPE). The models that are being compared are as follows: CNN (Gonzalez-Hernandez et al., 2018), RNN (Kollias & Zafeiriou, 2021), RCNN, SVR (Salama et al., 2021), and SSO-SVR (Agarwal & Om, 2021). Because the batch size of the data and the kernel function both have the potential to have a substantial impact on the performance of the model, these two parameters are the first ones to be determined. The results of the experiment are presented in Table 1 below for these two variables.

According to the findings in Table 1, the optimal configuration for the model is one in which the batch size is set to 128, and the sigmoid kernel function is utilized for the kernel analysis. As a direct consequence of this, the models that are being used in this investigation are being trained with this configuration. Last but not least, the results of the experiments are shown in Table 2 and Fig. 6, which can be accessed here.

The MAE provides a visual depiction of the model’s performance. The first row of data in Table 2 reveals that our model has the shortest MAE value, indicating that it outperforms multiple other models. The MAEs achieved by various deep learning models are lower than the SVR alone, indicating that the deep model performs better. As a result, it is preferable to present the depth model for feature extraction in this study. The variance in the genuine scale of different building projects, on the other hand, can cause MAE figures to vary greatly. As a result, in order to assess the model’s performance, measures such as MSE must be revisited. The MSE measure has the potential to quadruple the penalty for excessive mistakes. When compared to other models, our model has a distinct edge in MSE. This verifies our model’s superior predictive ability. However, the MSE measure continues to obtain extreme values and is not comprehensible. As a result, the MAPE indicator is also used in this work. This indicator is highly intuitive and allows you to see the difference between the real and forecasted values right away. We can observe from the experimental data that our model likewise performs well on this metric. As a result of merging the three measures, the experimental findings reveal that our model has a more accurate prediction function.

Table 1 Experimental results obtained with different parameter settings.

Batch size	Kernel functions	MAE	
32	Linear kernel function	1.5773	
64	Linear kernel function	1.5642	
128	Linear kernel function	0.9458	
32	Sigmoid kernel function	1.2327	
64	Sigmoid kernel function	0.7868	
128	Sigmoid kernel function	0.7345	

Table 2 Experimental results obtained from different models.

Index\Model	CNN	RNN	RCNN	SVR	SSO-SVR	Our model	
MAE	1.7567	1.4662	0.9377	1.4212	1.1393	0.7345	
MSE	0.3362	0.3257	0.3130	0.3419	0.3214	0.2936	
MAPE	9.4781	9.2346	8.9038	6.3572	5.8335	2.9343	

Figure 6 Comparison of experimental results.

Conclusion

The level of satisfaction that customers have with the functioning and other aspects of online platforms is directly proportional to the chance that those customers would make purchases through those platforms. You may therefore determine what individuals are going to buy based on how they are feeling when they are shopping online and using that information. When business owners do an excellent job of studying how customers feel about their purchases, it helps them promote, advertise, and sell more of their products. On the other hand, the improved precision of sentiment analysis can assist consumers in rapidly locating the fitness products that are their favorites, so saving them time that would otherwise be spent searching. Consequently, operational platforms have a lot to gain by developing the ability to anticipate how people are feeling. In this article, we propose an improved version of the SVR model for determining how people are feeling. The RCNN algorithm is used to extract features from the dataset by the enhanced SVR model. After gathering this information, the SVR is trained to make predictions about what customers will purchase. The findings of the studies indicate that the RCNN-optimized SVR model produces more reliable outcomes. Despite the fact that this study has improved our ability to forecast how customers will feel about shopping, there are still a few issues that need to be addressed. For instance, the migratory nature of the model needs to be investigated more and demonstrated. The algorithm makes use of shopping data obtained from an online fitness platform; however, additional research is required to see how accurately it can be transferred to different application scenarios. A deep learning approach is also utilized by the model in order to extract features. Although using this model makes it simpler to extract data features, running the method takes significantly more time because of it. These are the issues that need to be addressed in the further research that comes from this study. In light of the findings presented in the previous section of this paper, the following recommendations have been formulated with the goal of enhancing the service quality of online fitness platforms.

Improve information professionalism and control information release behavior

The online fitness platform’s primary function is to disseminate expert fitness information, and the high quality of this content is the primary selling point for its services. Manage the credibility of paid courses by enforcing tight rules and regulations from the perspective of the platform’s information providers; this will improve information quality; guarantee the credibility of platform information; and make use of appropriate rules and regulations. Make the most of the network’s potential as a social platform dedicated to the provision of fitness services by identifying and re-educating bloggers who produce inaccurate content, regulating the speech and behavior of its users, and transforming the network into a comprehensive social one. Management of the platform should be standardized, employees should have high levels of information literacy, an intelligent operation team should be assembled, and a robust and efficient system should be built on the foundation of a clear allocation of responsibilities.

Enhance the interactivity of engaging content to improve user immersion

Users can have a more thorough immersion experience by enhancing the usage of information technology, such as VR, to diversify and animate interaction. Customers may get the most out of their workout experience while also becoming exposed to cutting-edge new technologies if the two are combined. Adding additional community tools to increase user engagement with the platform system and bloggers. Promote user participation and emotional resonance through the use of topic circles and other techniques. To make text-only information more engaging, you may incorporate visual elements like pictures, sounds, and short videos; to emphasize key points and liven up the platform, you can use animation and doll-like features.

Enhance real-time engagement so that people experience instantaneous information exchange

Consumers’ expectations for instantaneous responses from their networks have increased alongside the development of more sophisticated network applications, but their patience for delays is wearing thin. If fitness is to continually improve the platform’s backend operations, allowing for faster page loads, information displays, and video playback, then the platform can better meet the needs of its users and provide them with more fluid engagement services. Accelerate the front-end sites and guarantee regular service usage in response to user requests. Better serve your customers by equipping your staff with the tools they need to respond to their questions and concerns in a timely manner, maintain their attention, and prevent them from going elsewhere.

Create more customized services, so that clients have a sense of special bespoke delight

Conduct research on the preferences of customers, investigate the needs of users, and improve the user-friendliness of your services. Consider the users’ unique routines, the pleasures they derive from their senses, and any other factors relevant to the users in order to comprehend the requirements of the users, to provide informational material and personalised services, and to enable the users to experience exclusive services. In the backend, a thorough user database is developed, and then a high-quality workout plan is crafted for the user, tailor-made to meet their requirements and based on their preferences as well as their goals. When gathering information on users, it is important to remember to protect their privacy and keep their information private. This will allow you to deliver information services that are more secure. Improve the functionality and standard of services provided by artificial intelligence while keeping in mind the adaptability and emotional transmission capabilities of these systems.

Supplemental Information

Supplemental Information 1 Google Analytics Customer Revenue Prediction dataset

Click here for additional data file.

Supplemental Information 2 Training model

Click here for additional data file.

Additional Information and Declarations

Competing Interests

Author Contributions

Data Availability

The authors declare there are no competing interests.

Mingkui Huo conceived and designed the experiments, performed the experiments, analyzed the data, performed the computation work, prepared figures and/or tables, authored or reviewed drafts of the article, and approved the final draft.

Jing Li conceived and designed the experiments, performed the experiments, analyzed the data, performed the computation work, prepared figures and/or tables, authored or reviewed drafts of the article, and approved the final draft.

The following information was supplied regarding data availability:

The data is available at Kaggle:

https://www.kaggle.com/datasets/colinpearse/ga-analytics-with-json-columns.

The raw measurements and code are available in the Supplementary Files.

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
