# Peer review of "A consumer emotion analysis system based on support vector regression model"

_PeerJ Computer Science, doi:10.7717/peerj-cs.1381_

## Round 0.1 · original submission · Major Revisions

Please revise your manuscript according to the obtained comments.

Reviewer 1 ·

Basic reporting

This article mines the online comments left by customers in order to assess the purchasing intentions of customers. The analysis of the emotional tendencies that customers have towards the purchase of products can assist online platforms in rapidly adjusting the quality and optimization direction of products or services, and it can also supply other consumers with buy comments and suggestions.

Experimental design

The results of the current experimental design and comparative analysis are sufficient, but there are also the following two problems:
1) The article does not provide a clear description of what each column in the dataset table in the experiment part signifies and the range of values it can take.
2) Can you perhaps explain the thought process that went into choosing the parameters that will be incorporated into the model?

Validity of the findings

1.The sixth and seventh sentences of the first paragraph of the introduction do not have the same level of authenticity as the rest of the paragraph. In order to make the essay more comprehensible, the author needs to work on improving the way the language is described.
2.The support vector regression algorithm was the primary algorithm that was utilized for the research. Either the article's summary or its introduction has to contain a condensed explanation of the rationale and the underlying principle behind the modification of the algorithm.
3.Consumer emotion recognition is the focus of the article, while emotion recognition based on speech is the subject of the reference [16]. Comparative analysis of other research would not add anything substantial to this one, thus include such studies would be pointless. It is recommended that the author present additional research on the emotional recognition of consumers.
4.The consumption mood prediction model has been updated to include the deep learning method known as the regression convolution neural network. The function of this model is described in Section 3.1 as being the extraction of features. Please explain the benefits of utilizing this model for the purpose of feature extraction.

Additional comments

This article mines the online comments left by customers in order to assess the purchasing intentions of customers. The analysis of the emotional tendencies that customers have towards the purchase of products can assist online platforms in rapidly adjusting the quality and optimization direction of products or services, and it can also supply other consumers with buy comments and suggestions. The significance of this work cannot be overstated. Despite this, the following issues with the article need to be resolved before it can be considered complete:
1.The sixth and seventh sentences of the first paragraph of the introduction do not have the same level of authenticity as the rest of the paragraph. In order to make the essay more comprehensible, the author needs to work on improving the way the language is described.
2.The support vector regression algorithm was the primary algorithm that was utilized for the research. Either the article's summary or its introduction has to contain a condensed explanation of the rationale and the underlying principle behind the modification of the algorithm.
3.Consumer emotion recognition is the focus of the article, while emotion recognition based on speech is the subject of the reference [16]. Comparative analysis of other research would not add anything substantial to this one, thus include such studies would be pointless. It is recommended that the author present additional research on the emotional recognition of consumers.
4.The consumption mood prediction model has been updated to include the deep learning method known as the regression convolution neural network. The function of this model is described in Section 3.1 as being the extraction of features. Please explain the benefits of utilizing this model for the purpose of feature extraction.
5.The text does not provide any additional information regarding the behaviors of the two relaxation variables that are presented in equation (4).
6.The flow of consumer sentiment prediction is provided in the article; however, the flow on its own is not sufficient to swiftly deploy the model. In addition to this, the particular steps required to put the concept into action should be detailed.
7.The article does not provide a clear description of what each column in the dataset table in the experiment part signifies and the range of values it can take.
8.Can you perhaps explain the thought process that went into choosing the parameters that will be incorporated into the model?

Reviewer 2 ·

Basic reporting

This essay, entitled "A Consumer Emphasis Analysis System Based on SVR Model," aims to do just that. The author refines the support vector regression model to increase the reliability of sentiment recognition from customers.

Experimental design

The procedure proposed by the author has been shown to be effective through experimentation.

Validity of the findings

Still, there is undoubtedly space for growth in the article's writing quality. The article might be improved by optimizing it; it includes a few typos that could be fixed which would improve its quality significantly.

Additional comments

1)Figure 1 is a representation of the rational behavior theory, and the article has to make it clear how this theory relates to the subject of this study, which is emotional recognition on the side of customers.
2)The mean absolute error, the mean standard error, and the mean absolute percentage error are the evaluation indicators that were used in the analysis of the experiment. What are the advantages of using a large number of indicators in order to assess the performance of the model that is being used for assessment?
3)The outcomes of an emotion forecast are utilized by the author in order to provide service recommendations for an online platform. On the other hand, the thought process that went into developing these recommendations ought to be given at the same time.
4)The data obtained from an online platform is analyzed in this study as part of the research on consumer sentiment that is conducted. What makes the information that is kept on the online platform unique are the identifying characteristics?
5)An improved version of the SVR algorithm was used in the consumer sentiment prediction model that was presented in this research. The model was proposed in this academic article. Moreover, the algorithm introduces RCNN for use in deep learning. There is a recommendation made regarding the mathematical principle of combining RCNN with SVR in the section of the article that discusses the model. The goal of this advice is to improve the accuracy of the model's predictions.
6)Utilizing a flow chart that illustrates the various processes involved in the process of forecasting the feelings of customers will help make the process more understandable overall.
7)There is opportunity for further improvement in terms of the overall quality of the English language usage throughout the article. There are certain assertions that aren't very understandable at all. In order to improve the article's overall quality and make it more useful to readers, its author has to polish their work even further.

---

## Round 0.2 · accepted · Accept

Now this article can be accepted in its current form.

Reviewer 1 ·

Basic reporting

No more comments

Experimental design

No more comments

Validity of the findings

No more comments

Additional comments

I appreciate the efforts from the authors in revising the manuscript. The revision has fully addressed all my concerns. The response letter and the edited paragraphs help clarify on the algorithmic details and performance of the proposed method. I would like to recommend acceptance of the paper in the current form.

Reviewer 2 ·

Basic reporting

The effective means to stimulate economic growth is to enhance consumers' consumption capacity. Because many consumers have different consumption habits, they will pay different attention to products. Even the same consumer will have different shopping experiences when buying the same product at different times. By mining the online comments of consumers on the online fitness platform, we can find the characteristics of fitness projects that consumers care about. Analyzing consumers' emotional tendencies towards the characteristics of fitness programs will help online fitness platforms adjust the quality and service direction of fitness programs in a timely manner. At the same time, it can also provide purchase advice and suggestions for other consumers. Based on this goal, this study uses an optimized support vector regression (SVR) model to build a consumer sentiment analysis system, so as to predict the consumer's willingness to pay.

Experimental design

The optimized SVR model uses Region Convolution Neural Network (RCNN) to extract features from the dataset, and uses feature data to train the SVR model. The experimental results show that the SVR model optimized by RCNN is more accurate. The improvement of the accuracy of consumer sentiment analysis can accurately help businesses promote and publicize, and increase sales. On the other hand, the increase in the accuracy of emotion analysis can also help users quickly locate their favorite fitness projects, saving browsing time.

Validity of the findings

To sum up, the emotional analysis system for consumers in this paper has good practical value.

Additional comments

I think this paper can be accepted in current version.